# Forecasting large hail and lightning using additive logistic regression models and the ECMWF reforecasts

**Francesco Battaglioli**[1,2]**, Pieter Groenemeijer**[1,3]**, Ivan Tsonevsky**[4]**, and Tomàs Púčik**[3]

[1]European Severe Storms Laboratory e.V. (ESSL), 82234 Wessling, Germany
[2]Institut für Meteorologie, Freie Univerisität Berlin, 12165 Berlin, Germany
[3]European Severe Storms Laboratory (ESSL) – Science and Training, 2700 Wiener Neustadt, Austria
[4]European Centre for Medium Range Weather Forecasts (ECMWF), Reading, RG2 9AX, United Kingdom

**Correspondence:** Francesco Battaglioli (francesco.battaglioli@essl.org)

**Abstract.** Additive logistic regression models for lightning ($AR_{lig}$) and large hail ($AR_{hail}$) were developed using convective parameters from the ERA5 reanalysis, hail reports from the European Severe Weather Database (ESWD), and lightning observations from the Met Office Arrival Time Difference network (ATDnet). The models yield the probability of lightning and large hail in a given timeframe over a particular grid point. To explore the value of this approach to medium-range forecasting, the models were applied to the European Centre for Medium Range Weather Forecasts (ECMWF) reforecasts to reconstruct probabilistic lightning and large hail forecasts for 11 ensemble members, from 2008 to 2019 and for lead times up to 228 h. The lightning and large hail models were based on different predictor parameters: most unstable convective available potential energy (CAPE), 925–500 hPa bulk shear, mixed layer mixing ratio, wet bulb zero height (for large hail), most unstable lifted index, mean relative humidity between 850 and 500 hPa, 1 hourly accumulated convective precipitation and specific humidity at 925 hPa (for lightning). First, we compared the lightning and hail ensemble forecasts for different lead times with observed lightning and hail focusing on a recent hail outbreak. Second, we evaluated the predictive skill of the model as a function of forecast lead time using the area under the ROC curve (AUC) as a validation score. This analysis showed that $AR_{hail}$ has a very high predictive skill (AUC > 0.95) for a lead time up to 60 h. $AR_{hail}$ retains a high predictive skill even for extended forecasts (AUC = 0.86 at 180 h lead time). Although $AR_{lig}$ exhibits a lower predictive skill than $AR_{hail}$, lightning forecasts are also skilful both in the short term (AUC = 0.92 at 60 h) and in the medium range (AUC = 0.82 at 180 h). Finally, we compared the performance of the 4-dimensional hail model with that of composite parameters such as the significant hail parameter (SHP) or the product of CAPE and the 925–500 hPa bulk shear (CAPESHEAR). Results show that $AR_{hail}$ outperforms CAPESHEAR at all lead times and SHP at short-to-medium lead times. These findings suggests that the combination of additive logistic regression models and ECMWF ensemble forecasts can create highly skilful medium-range hail and lightning forecasts for Europe.

## 1 Introduction

Lightning and large hail are responsible every year for large societal and economic impacts in Europe. While hail is mostly responsible for high economic losses with single events causing more than USD 1 billion in damage (Púčik et al., 2019), lightning is responsible for hundreds of injuries and an average of 64 fatalities per year in Europe only (Kühne et al., 2023). Such extensive economic and societal impacts have sparked research on different aspects of these events ranging from the development of climatologies (hail: Cintineo et al., 2012; Cecil and Blankenship, 2012; Bang and Cecil, 2019; Púčik et al., 2019; Taszarek et al., 2020a; Fluck et al., 2021; Murillo et al., 2021, lightning: Feudale et al., 2013; Wapler, 2013; Taszarek et al., 2019; Enno et al., 2020), to nowcasting (hail: Ryzhkov et al., 2013; Ortega et al., 2016; Nisi et al., 2020; Schmidt, 2020, lightning: Mostajabi et al., 2019; Cintineo et al., 2022; Leinonen et al., 2022)

and forecasting (hail: Jewell and Brimelow, 2009; Adams-Selin and Ziegler, 2016; Czernecki et al., 2019; Allen et al., 2020, lightning: McCaul et al., 2009; Zepka et al., 2014; Dafis et al., 2018; Geng et al., 2021). Lightning can be forecasted based on the pre-convective environment using the ingredient-based methodology (Johns and Doswell, 1992) of deep-moist convection for which convective storms require three ingredients to form: conditional instability, moisture and lift. Although all convective storms require these three ingredients, not all of them produce lightning. Van Den Broeke et al. (2005) found that sufficient instability must exist in the lower mixed phase region of the cloud for lightning to occur. Their findings have been used to create a physical-based parameter for lightning prediction at the Storm Prediction Center (Bright et al., 2005). Westermayer et al. (2017) have found that the relative frequency of lightning increases as convective available potential energy (CAPE) grows between 0 and $200 \, \mathrm{J \, kg^{-1}}$, levelling off after that. The frequency only increases further once the mid-tropospheric relative humidity increases and/or Convective Inhibition (CIN) decreases. Another approach to lightning forecasting is using explicit simulations of a thunderstorm updraft and its microphysics. This is the case for the lightning potential index (LPI) developed by Yair et al. (2010) and applied in a number of convection-allowing models (Lagasio et al., 2017; Brisson et al., 2021; Uhlířová et al., 2022). For global, lower-resolution models, a combination of pre-convective environment and cloud properties have been used to parameterize lightning (Lopez, 2016). Hail forecasting techniques have been widely explored in the United States both with the development of hail models based on the explicit simulation of hail growth mechanisms in a thunderstorm, e.g. HAIL-CAST (Brimelow et al., 2002; Jewell and Brimelow, 2009; Adams-Selin and Ziegler, 2016), but also with the use of convection-allowing models (Gallo et al., 2016, 2018) and convection-allowing ensembles (Loken et al., 2020). In Europe, hail research has mostly concentrated on (semi-) automatic nowcasting techniques using radar data (Martius et al., 2015). Although successful medium-range hail forecasts can mitigate the impacts of hail and have been shown to be skilful in the United States (Lepore et al., 2017, 2018), little attention has been given to the development of specific products for medium-range hail forecasting in Europe. At such lead times, convection-allowing models are not available, but an ingredients-based approach (Doswell et al., 1996) can be used, whereby forecasters look for the simultaneous presence of prerequisites needed to sustain hailstorms, such as instability, moisture, lift, and a minimum amount of wind shear. Composite parameters such as the large hail parameter (LHP; Johnson and Sugden, 2014), the supercell composite parameter (SCP; Thompson et al., 2004) and the significant hail parameter (SHP; http://www.spc.noaa.gov/exper/mesoanalysis/help/help_sigh.html, last access: 22 February 2023) can help by combining several ingredients into one number describing whether the environmental conditions are

supportive for large hail and can be used for medium-range forecasting. SHP calculated from the Global Ensemble Forecast System (GEFS) yielded skilful probabilistic hail forecasts up to 12 d in advance in the United States (Gensini and Tippett, 2019). Although composite parameters such as SHP perform well across the United States, climatologies based on them perform worse across regions where the parameter has not been specifically developed (e.g. Europe, as found by Taszarek et al., 2020b). Battaglioli et al. (2023) showed that the additive logistic regression models (AR-CHaMo) for large hail occurrence have a better predictive skill than SHP and can result in a more realistic climatological distribution of hail occurrence across Europe. Building upon these findings, in this study we leverage an ensemble approach in conjunction with the lightning and hail models from Battaglioli et al. (2023) to yield probabilistic lightning and hail forecasts with the ultimate goal of improving medium-range forecasting of these hazards in Europe.

## 2  Data

### 2.1  AR-CHaMo lightning and hail models

The AR-CHaMo models were developed by Rädler et al. (2018) and improved by Battaglioli et al. (2023). The models were trained using lightning observations from the Arrival Time Difference Network (ATDNet, Anderson and Klugmann 2014; Enno et al., 2020), hail reports from the European Severe Weather Database (ESWD; Dotzek et al., 2009; Groenemeijer et al., 2017) and convective parameters from the ERA5 reanalysis (Hersbach et al., 2020). The model training area for large hail was limited to Central Europe (Fig. 1a) where the reporting of severe weather in the ESWD is the highest (Groenemeijer and Kühne, 2014; Taszarek et al., 2020a). This was done to limit sampling of situations in our training dataset that could have affected the model development, namely when hail occurred but was not reported to the ESWD. The lightning model training region covered a much larger area (Fig. 1b) since the detection efficiency of ATDnet is homogeneous across the European domain. Lightning and hail reports were gridded on a $0.25° \times 0.25°$ grid, the same horizontal resolution of the ERA5 reanalysis. In addition, the time of reports was rounded down to the nearest full hour. This allowed us to best sample the pre-convective environment associated with each lightning observation or hail report, as done by Rädler et al. (2018) and Taszarek et al. (2021b). The AR-CHaMo models use logistic regression to assign a probability of hazard occurrence as a function of reanalysis-derived predictor parameters. AR-CHaMo separately predicts the probability of thunderstorm formation and the probability of large hail ($\geq 2 \, \mathrm{cm}$) given that a thunderstorm formed. The probability of large hail ($P_{\mathrm{hail} > 2 \, \mathrm{cm}}$) is a product of these two components, as shown in Eq. (1):

$$P_{\mathrm{hail} > 2 \, \mathrm{cm}} = P_{\mathrm{lightning}} \cdot P_{\mathrm{hail} > 2 \, \mathrm{cm}|\mathrm{lightning}} \qquad (1)$$

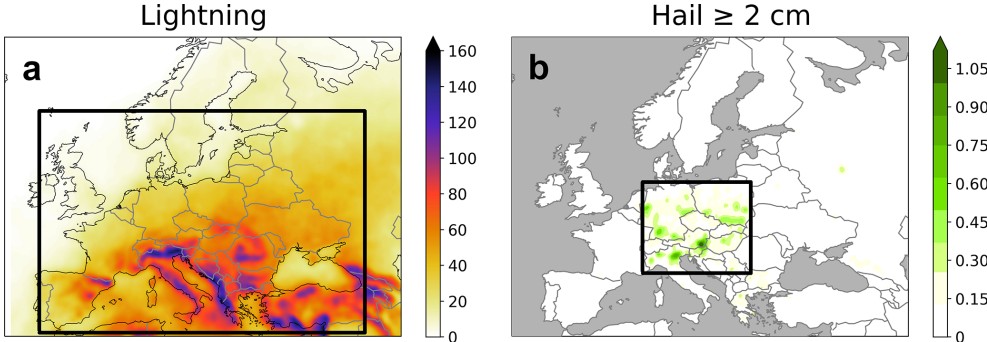

**Figure 1.** Annual mean distribution of lightning **(a)** and hail $\geq 2$ cm **(b)** for the period 2008–2019. The black squares in **(a)** and **(b)** indicate the training regions for lightning (34.5–63.5° N, −9.0–46.0° W) and hail $\geq 2$ cm (45.0–54.0° N, 5.0–22.0° W). Adapted from Battaglioli et al. (2023).

**Table 1.** Selected predictor parameters for the original AR-CHaMo models. Acronyms are explained in Appendix D.

| Model | Lightning | Hail $\geq 2$ cm |
|---|---|---|
| Predictors | MU_LI (K) | MU500_CAPE-10° (J kg$^{-1}$) |
| | RH_500–850 hPa (%) | EFF_MU_BS (m s$^{-1}$) |
| | 1 h Acc. conv. precip. (kg m$^{-2}$) | MU MIXR (g kg$^{-1}$) |
| | MU_MIXR (g kg$^{-1}$) | 0° height (m) |
| | Land–sea mask | |

Different predictor parameters from the ERA5 reanalysis were selected for the lightning and the hail model following a model selection procedure on the basis of an ingredients-based approach (Doswell et al., 1996), the deviance explained (Wood, 2017) and the Bayesian information criterion (BIC, Schwarz, 1978) score. Out of 172 available parameters from the ERA5 reanalysis, the model selection procedure with deviance explained (the higher, the better) and BIC (the lower, the better) yielded a 5-dimensional lightning model and a 4-dimensional conditional hail $\geq 2$ cm model. More details on the model selection procedure can be found in Battaglioli et al. (2023) while the final model predictors for the lightning and the conditional hail models are listed in Table 1.

## 2.2 ECMWF forecast data

Forecast data across Europe were obtained from the European Centre for Medium Range Weather Forecasts (ECMWF) reforecasts (Vitart, 2013) for the period 2008–2019. ECMWF reforecasts used in this study consist of 10 perturbed ensemble members and one control forecast run every Monday and Thursday. These reforecasts, unlike reanalysis, always use the operational version of the ECMWF Integrated Forecasting System (IFS), covering two IFS cycles: 46r1 until 30 June 2020 and 47r1 afterwards. Such change in IFS cycle is, however, not expected to have caused

discontinuities in the calculation of convective parameters (e.g. in CAPE) since the convection scheme was not modified from 46r1 to 47r1. Reforecast data were extracted at a spatial resolution of 0.2° × 0.2° and 6 hourly temporal resolution. The forecast parameters downloaded from reforecasts were temperature, specific humidity, u and v wind components at the 1000, 850, 700 and 500 hPa pressure levels. They were used to compute convection-related parameters such as mixing ratio, mid-level relative humidity and 925–500 hPa bulk shear (deep layer shear). Composite parameters, such as CAPESHEAR (a product of $\sqrt{\text{CAPE}}$ and deep layer shear) and SHP were also calculated while CAPE for the most-unstable parcel (MUCAPE) and convective precipitation were obtained directly from the IFS.

## 2.3 Lightning observations and hail reports

Lightning data from ATDnet were used to verify lightning forecasts from AR-CHaMo. This network can detect lightning flashes even at large distances from a sensor and enables verification of the lightning forecasts across a broad region covering most of Europe. Data were obtained for the period 2008–2019, gridded on a 0.2° × 0.2° grid to allow for direct comparison with ECMWF reforecasts, and organized as a binary field indicating whether a lightning case occurred (1 = yes, 0 = no). A lightning case was defined as a 1 h period with at least two lightning strikes per grid box. Following Rädler et al. (2018), single detections were ignored as they might be isolated measurement errors rather than lightning. Hail reports were obtained from the ESWD and used to verify the hail forecasts. As per Rädler et al. (2018), reports with an excessive time uncertainty (> 1 h) were discarded from the verification dataset. In total, 10 872 890 lightning observations and 5493 reports of large hail were considered within the verification dataset.

## 3   Model development and adaptation

The models for lightning and large hail developed by Battaglioli et al. (2023) were based on the predictors from the ERA5 reanalysis listed in Table 1. Owing to the limited vertical resolution of the ECMWF reforecast data, some of the original AR-CHaMo predictors (e.g. MU500_CAPE-10°) could not be calculated using the ECMWF reforecasts. For this reason, the models were adapted using predictors as close as possible to their original versions (e.g. MU_CAPE was chosen instead of MU500_CAPE-10°). The models were then trained again using the adapted predictors (shown in Table 2) using ERA5 reanalysis, lightning and hail observations. The models were not retrained using the ECMWF reforecasts owing to the nature of the dataset: reforecasts are initialized only twice a week and the representation of the environmental conditions in proximity of a report is dependent on the forecast lead time (the closer to the initialization time, the better). By training a single model for all lead times, if, for instance, two hail reports occurred on Tuesday at 12:00 UTC and on Wednesday at 18:00 UTC, it would not be possible to qualitatively compare the environmental conditions at these two time steps. This is because for a forecast initialized on Tuesday at 00:00 UTC, the environmental conditions associated with the Wednesday at 18:00 UTC report would be subject to larger uncertainty due to the larger lead time. The models based on ERA5 assign a probability of hazard occurrence as a function of the reanalysis-derived predictor parameters listed in Table 2 to any location in a $0.25° \times 0.25°$ grid and 1 hourly intervals. Given the different spatial (ERA5: $0.25° \times 0.25°$, ECMWF reforecast: $0.20° \times 0.20°$) and temporal (1 and 6 h) resolution between ERA5 and ECMWF reforecasts, the probabilities had to be readapted. To account for the different spatial resolution of ECMWF reforecasts, the ERA5 $P_{\text{hazard(ERA5)}}$ probabilities were adapted following Eq. (2):

$$P_{\text{hazard(ECMWF)}} = 1 - (1 - P_{\text{hazard(ERA5)}})^{A_{\text{ECMWF}}/A_{\text{ERA5}}}, \quad (2)$$

where $A_{\text{ECMWF}}$ is the area of an ECMWF reforecast grid box ($\text{km}^2$), $A_{\text{ERA5}}$ is the area of an ERA5 grid box ($\text{km}^2$), and $P_{\text{hazard(ECMWF)}}$ is the adapted probability using an ECMWF reforecast grid (%).

Given that ECMWF reforecasts are available every 6 h, lightning and large hail probabilities could be calculated only at four different hourly time steps during the day (00:00–01:00, 06:00–07:00, 12:00–13:00, 18:00–19:00 UTC). Using only four hourly time steps would significantly limit the amount of large hail reports to work with for verification. Therefore, probabilities were upscaled to 3 h intervals (00:00–03:00, 06:00–09:00, 12:00–15:00, 18:00–21:00 UTC) following Eq. (3), based on the assumption that each hour probability is independent of that in another hour:

$$P_{\text{hazard(3 hourly)}} = 1 - (1 - P_{\text{hazard(1 hourly)}})^3 \qquad (3)$$

With these adaptations, we applied the AR-CHaMo based on ERA5 to ECMWF reforecasts yielding 3 hourly ensemble lightning and hail probabilistic forecasts at $0.20° \times 0.20°$ spatial resolution for the period 2008–2019 and for the whole of Europe.

## 4   Application of hail and lightning models to a case study: 15 June 2019

### 4.1   Ensemble forecasts

We first tested the AR-CHaMo-based lightning and hail forecasts on a case study. On 15 June 2019, severe storms with large hail occurred in eastern Germany, western Poland and Czechia. More than 20 hail reports were submitted for the period 12:00–15:00 UTC with several hail reports exceeding 5 cm in diameter. A second region of interest on the day was south-eastern France where very large hail was also reported (8 cm near Grenoble). We evaluated the performance of the ensemble forecasts depending on the lead time by considering ECMWF reforecasts initialized at three different lead times ($t - 12$ h, $t - 108$ h and $t - 180$ h) ahead of the event. To allow for a comparison with hail forecast products from the US Storm Prediction Centre (SPC), probabilities were upscaled using an adapted version of Eq. (2) to yield a probability of hail $\geq 2$ cm occurrence in a radius of 25 miles of a point, approximately 40 km. For hail forecasts for 15 June 2019 at 12:00 UTC, initialized at 00:00 UTC on the same day, a strong agreement between all ensemble members was present in identifying the Germany–Poland–Czechia region and south-eastern France as the areas with highest hail potential ($P_{\text{hail} > 2\,\text{cm}} > 30$ %) (Fig. 2). To compare the AR-CHaMo forecast with that of existing composite parameters, we produced probabilistic hail forecasts for the same time step and initialization time based on two 1-dimensional logistic models trained using SHP (Fig. 3) and CAPESHEAR (Fig. 4). The SHP model is in agreement regarding the Germany–Poland–Czechia and south-eastern France regions, but compared to AR-CHaMo, yields high hail probabilities also across regions where no hail was reported, e.g. the Balkans and Eastern Europe. The CAPESHEAR model, on the other hand, identifies well the south-eastern France region but places the highest probability of hail across northern Germany and northern Poland away from the highest density of hail reports to the south. Similarly to the hail forecasts, there was a strong agreement in the lightning ones among the ensemble members at 12 h lead time (Fig. 5). A high probability of lightning ($P_{\text{lightning}} > 80$ %) was present across the German–Polish border and south-eastern France. Between 12:00 and 15:00 UTC, widespread lightning activity occurred across a broad region extending from northern Germany and Denmark all the way south to southern Austria, verifying the lightning forecast in these areas. In addition to these regions, high lightning probabilities were

**Table 2.** Chosen predictor parameters for the regression models to be applied to the ECMWF reforecasts. Predictors highlighted with an asterisk are the ones that have been changed compared to the original ERA5 ones listed in Table 1.

| Model | Lightning | Hail $\geq 2$ cm |
|---|---|---|
| Predictors | MU_LI (K) | MU_CAPE (J kg$^{-1}$)* |
| | RH_500–850 hPa (%) | Deep-layer shear (m s$^{-1}$)* |
| | 1 h Acc. conv. precip. (kg m$^{-2}$) | Specific humidity at 925 hPa (g kg$^{-1}$)* |
| | Specific humidity at 925 hPa (g kg$^{-1}$)* | Wet bulb zero height (m) |

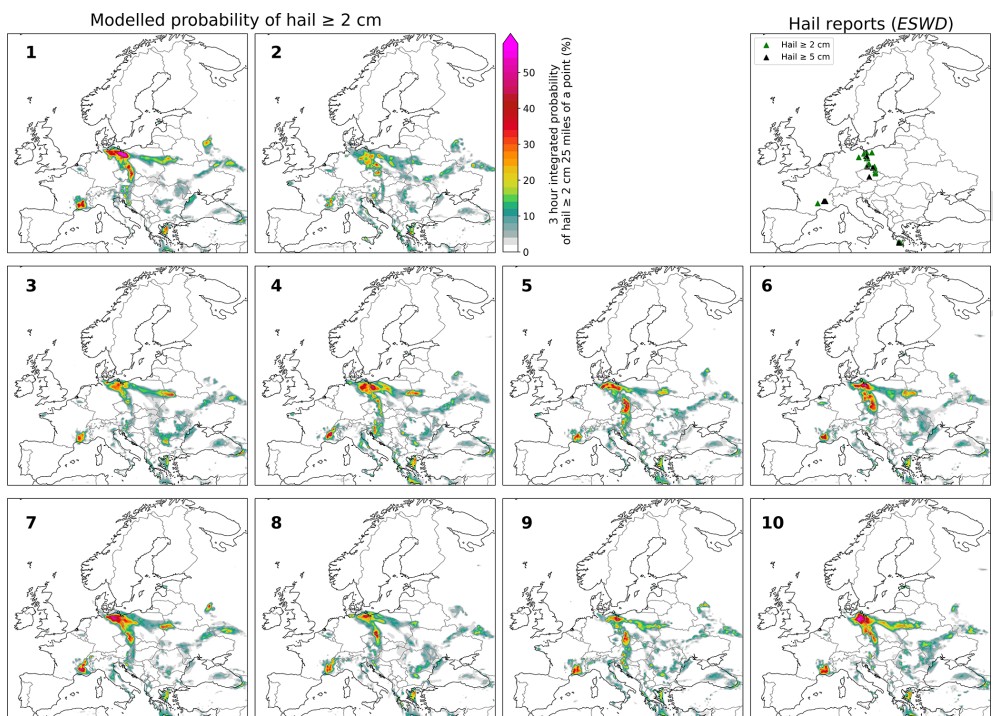

**Figure 2.** Probabilistic forecast of hail $\geq 2$ cm occurrence on 15 June 2019 at 12:00 UTC (initialized on 15 June 2019 at 00:00 UTC) for the individual ensemble members. Hail reports between 12:00 and 15:00 UTC are shown as triangles (green for hail $\geq 2$ cm but $\leq 5$ cm, black for hail $\geq 5$ cm) in the top-right panel.

found along a corridor extending from Ukraine into Russia. Widespread lightning activity was also observed here. An underestimation of the lightning activity was present across western Türkiye, where thunderstorms occurred despite low CAPE ($< 100$ J kg$^{-1}$; see sounding in Appendix A).

## 4.2 Physical interpretation of model forecasts

Forecast soundings and hodographs can show how individual parameters influenced the probability of lightning or hail simulated by AR-CHaMo. Two regions, central Ukraine and central Italy, showed respectively high and low probabilities of lightning (Fig. 6). While moderate buoyancy (CAPE $> 1000$ J kg$^{-1}$) was present in both regions, the main difference in the two profiles was the mid-level relative humidity, significantly lower in the Italian sounding. Here a deep, dry layer in the mid-troposphere could have resulted in dry air entrainment of updrafts that inhibited convec-

tive initiation (Rädler et al., 2018; Poręba et al., 2022). Although this analysis provides a physical explanation of the different lightning probabilities, it is important to note that here we considered vertical profiles from the two deterministic runs only, while the profiles of the single ensemble members could differ significantly, for instance in the representation of mid-level moisture. The comparison between hail and lightning forecasts highlights the ability of the model to distinguish between areas with lightning potential and with hail potential. In a region extending from Romania into Ukraine and Russia, low probabilities of large hail ($P_{\text{hail} > 2\,\text{cm}} < 10\,\%$) were found (Fig. 5) alongside high lightning probabilities ($P_{\text{lightning}} > 70\,\%$). On the other hand, across eastern Germany the probabilities of lightning ($P_{\text{lightning}} > 70\,\%$) and hail ($P_{\text{hail} > 2\,\text{cm}} > 25\,\%$) were both relatively high. While high CAPE was present both across Ukraine and eastern Germany, the deep-layer shear

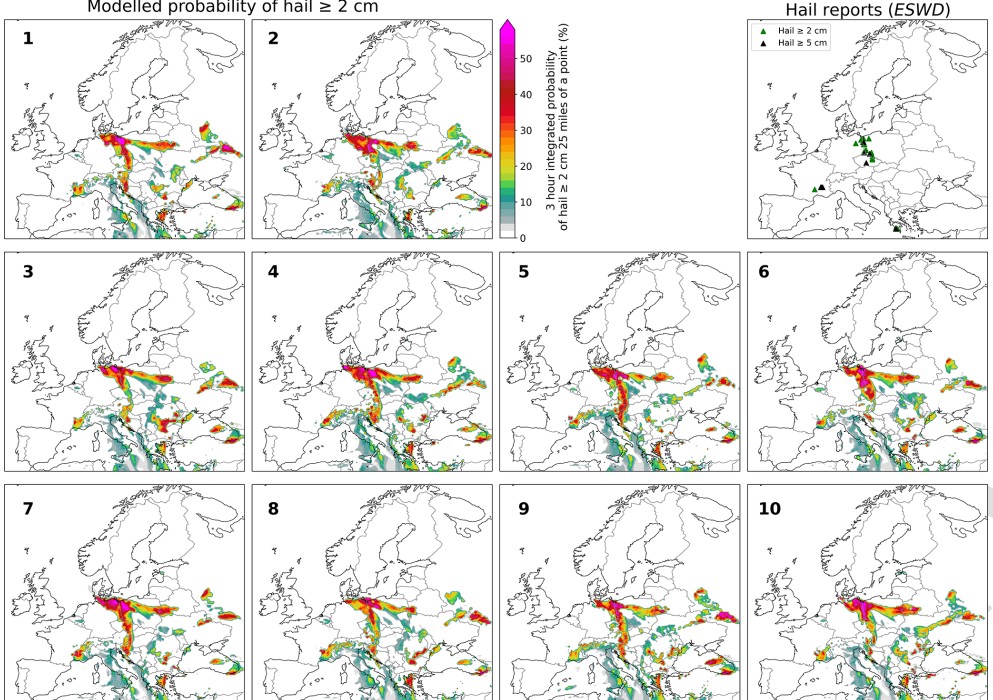

**Figure 3.** Probabilistic ensemble forecast of hail occurrence based on a 1-dimensional logistic SHP model for 15 June 2019 at 12:00 UTC (initialized on 15 June 2019 at 00:00 UTC). Hail reports between 12:00 and 15:00 UTC are shown as triangles (green for hail ≥ 2 cm but ≤ 5 cm, black for hail ≥ 5 cm) in the top-right panel.

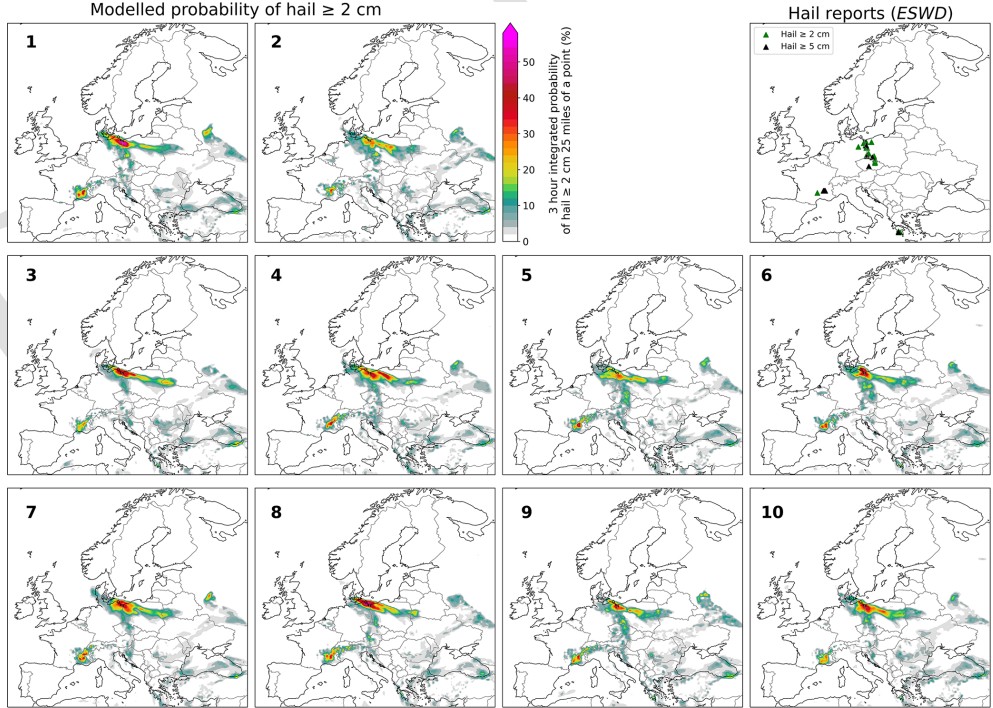

**Figure 4.** Probabilistic ensemble forecast of hail occurrence based on a 1-dimensional logistic CAPESHEAR model for 15 June 2019 at 12:00 UTC (initialized on 15 June 2019 at 00:00 UTC). Hail reports between 12:00 and 15:00 UTC are shown respectively as triangles (green for hail ≥ 2 cm but ≤ 5 cm, black for hail ≥ 5 cm) in the top-right panel.

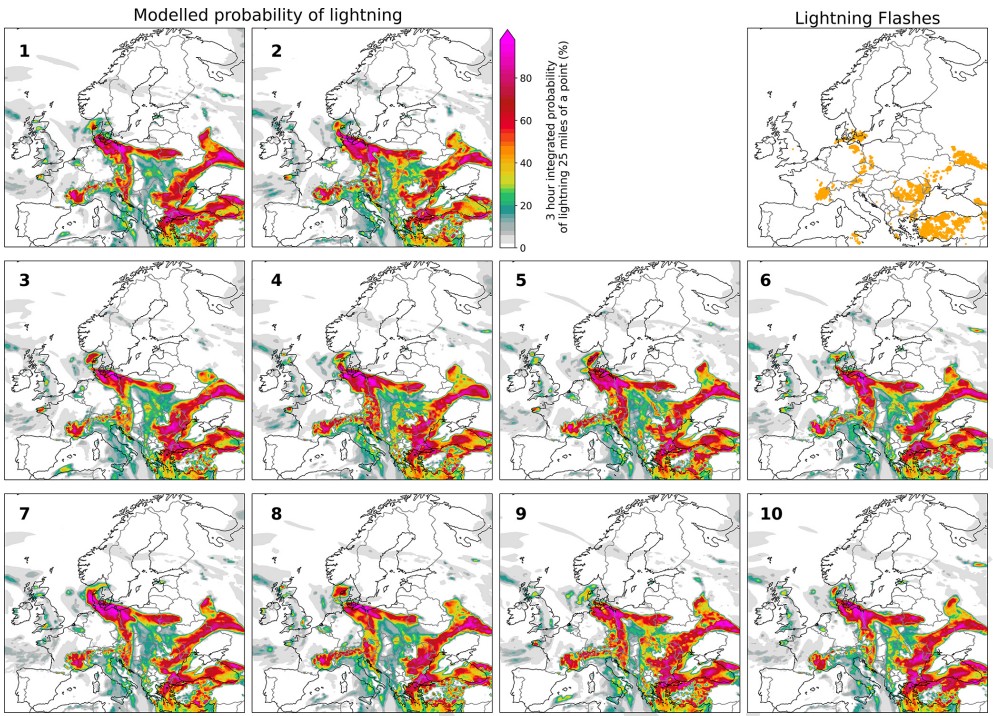

**Figure 5.** Probabilistic forecast of lightning occurrence on 15 June 2019 at 12:00 UTC (initialized on 15 June 2019 at 00:00 UTC) for the individual ensemble members. Lightning observations between 12:00 and 15:00 UTC are shown as orange dots in the top-right panel.

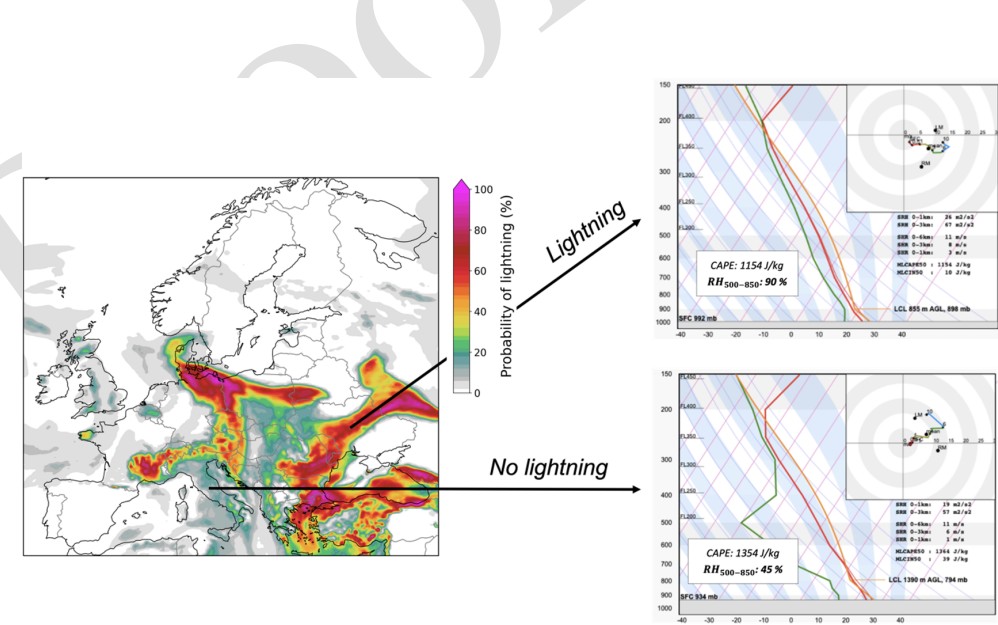

**Figure 6.** Ensemble mean probabilistic forecast of lightning occurrence on 15 June 2019 at 12:00 UTC (initialized on 15 June 2019 at 00:00 UTC). Forecast soundings from the ECMWF deterministic run are shown for two locations: central Ukraine and central Italy. Corresponding CAPE and relative humidity between 500 and 850 hPa ($RH_{500-850}$) values are also shown. Hodographs are plotted in red (0–1 km), yellow (1–3 km), green (3–6 km) and blue (6–10 km) for the respective height intervals.

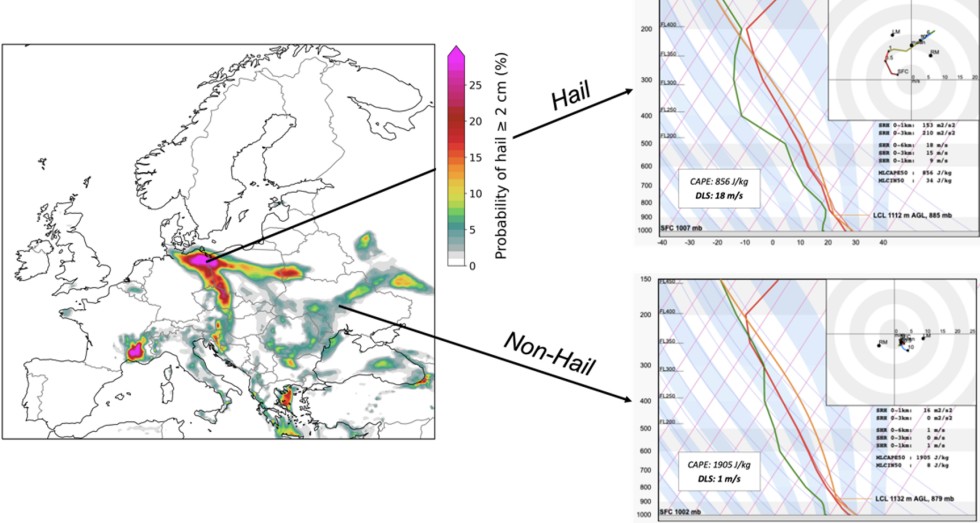

**Figure 7.** Ensemble mean probabilistic forecast of hail ≥ 2 cm occurrence on 15 June 2019 at 12:00 UTC (initialized on 15 June 2019 at 00:00 UTC). Forecast sounding from the ECMWF deterministic run are shown for two locations: eastern Germany and central Ukraine. Corresponding CAPE and deep-layer shear are also shown. Hodographs are plotted in red (0–1 km), yellow (1–3 km), green (3–6 km) and blue (6–10 km) for the respective height intervals.

was much stronger over Germany (Fig. 7). Stronger shear is associated with a higher likelihood of large hail, because it promotes storm organization (Thompson et al., 2012; Johnson and Sugden, 2014; Púčik et al., 2015; Taszarek et al., 2020b; Kumjian et al., 2021). This was well reflected by AR-CHaMo that showed a higher hail probability across eastern Germany in correspondence with the strongest shear.

### 4.3   Forecast dependence on lead time

We next investigated how lead time affects forecast performance by comparing the hail forecast for 15 June 2019 at 12:00 UTC initialized on 11 June 2019 at 00:00 UTC ($t - 108$ h, Fig. 8) with the $t - 12$ h forecast (Fig. 2). Despite the increased lead time, the ensemble members were still in good agreement regarding the location of the highest hail probabilities across the German–Polish–Czech area even 4 d in advance. The largest difference compared to the $t - 12$ h forecast was found across south-eastern France where most members showed no probability of large hail occurrence.

The reason for the large spread in the hail probabilities was a large spread in the predicted CAPE (Fig. 10), which was much lower for shorter lead times (Fig. 9). While the $t - 12$ h forecast highlights a localized spot of high CAPE across south-eastern France, this region of enhanced buoyancy was not represented by most members in the $t - 108$ h forecast. Forecasts of deep-layer shear (not shown) for the two lead times did not show differences that might impact the forecast across south-eastern France. As the lead time and spread among the forecast members increased, the probabilities decreased (Fig. 11). A similar dependence on the lead time

was observed for the 1-dimensional logistic models based on SHP and CAPESHEAR (Appendix B). Despite the increased spread compared to the $t - 12$ h forecast, the $t - 108$ h hail forecast had a good predictive skill across the German–Polish–Czech area with a widespread 10 %–15 % probability of occurrence, which could be clearly distinguished from surrounding areas with very low probabilities. The $t - 108$ h lightning forecast (Appendix C) was also in good agreement with the $t - 12$ h forecast, although lower confidence was found across Eastern Europe where uncertainty regarding the location of the initiation boundary existed. The spread increased further in the $t - 180$ h forecast for both hail and lightning (Fig. 11). For this case study we showed that the logistic model applied to the ECMWF reforecasts provided a skilful forecast at least up to 108 h in advance for both hail and lightning. The loss of predictive skill from 12 to 108 h for the hail model was mostly caused by an increasing uncertainty in the CAPE forecast with increasing lead time.

## 5   Model evaluation

In order to provide a more general evaluation of the skill of the model not tied to a single case study, the ensemble mean probabilistic forecasts were systematically verified at four different times during the day (00:00, 06:00, 12:00, and 18:00 UTC) against lightning observations from ATDnet in the 1 h following the forecast time (00:00–01:00, 06:00–07:00, 12:00–13:00, and 18:00–19:00 UTC) and hail reports from the ESWD in the 3 h following the forecast time (00:00–03:00, 06:00–09:00 12:00–15:00, and 18:00–

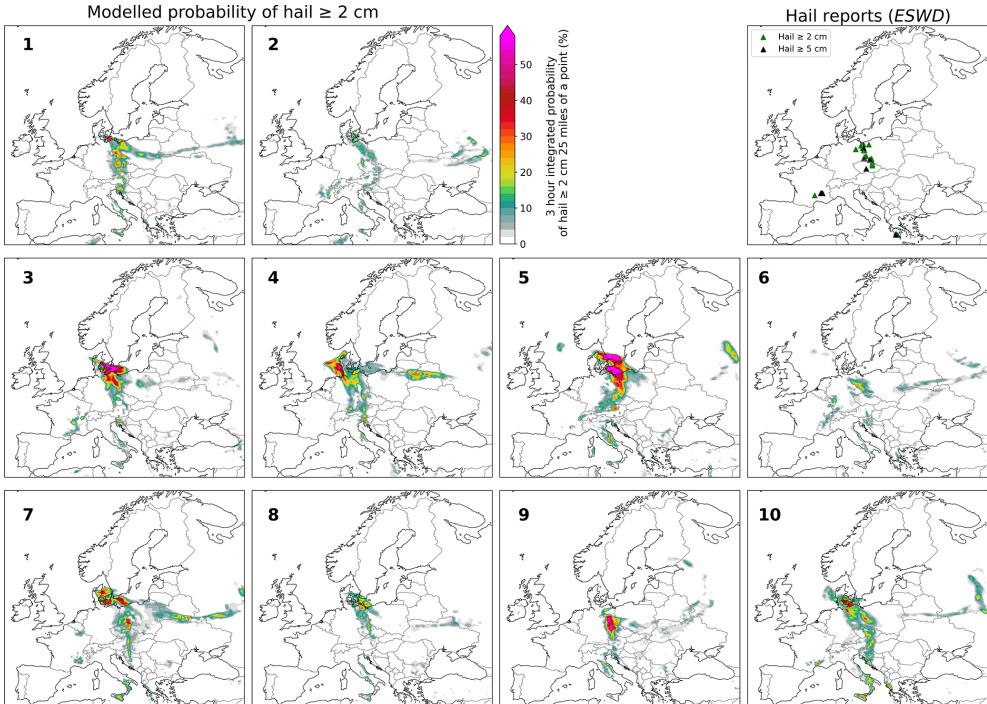

**Figure 8.** As in Fig. 2 but for a forecast initialized on 11 June 2019 at 00:00 UTC.

21:00 UTC). While for lightning the full European domain was used to verify lightning forecasts owing to the homogeneous detection efficiency of ATDnet across the domain, hail forecasts were only verified against ESWD hail reports in Central Europe (Fig. 1b). This reduced area was chosen because of a lack of reports from some regions, such as southeastern Europe (Rädler et al., 2018). We used the receiver operating characteristic (ROC) curves for lead times from 12 to 228 h (at 24 h time steps) to investigate changes in forecast skill of the lightning and hail models, similarly to Tsonevsky et al. (2018). The area under the ROC curve (AUC, Wilks, 1995) measures the ability of the hail (lightning) model to discriminate between hail (lightning) and non-hail (non-lightning) situations. The AUC scores for the hail model were higher than the lightning model ones at all lead times (Fig. 12). The hail model exhibited a very high predictive skill: $AUC > 0.95$ for lead times up to 60 h. Although the predictive skill decreased with increasing lead time, a high AUC was still found for lead times up to 108 h. A relatively strong discontinuity in predictive skill was found between 108 (0.920) and 132 h (0.873) and after 180 h. The lightning model exhibited an almost linear decrease of predictive skill with increasing lead time. The performance of $AR_{hail}$ was compared with that of 1-dimensional logistic models trained using CAPESHEAR and SHP as predictors and the same training dataset as for $AR_{hail}$. In Fig. 11, we display the AUC scores for $AR_{hail}$ and for the models based on the two composite parameters as a function of the forecast lead time. To quantify the uncertainty in the AUC for the different models,

we performed a 1000-member block bootstrap procedure, as done by Hamill et al. (2018), which allowed us to determine the corresponding 95 % confidence intervals (Fig. 13). Comparing the AUC scores of the probabilistic forecasts based on the 1-dimensional CAPESHEAR model and those of $AR_{hail}$, we concluded that $AR_{hail}$ outperformed CAPESHEAR at all lead times. Compared to SHP, $AR_{hail}$ had a higher performance at short-to-medium range (up to 60–84 h) while with increasing lead time the two metrics became comparable in terms of predictive skill. It is hypothesized that the increase in the uncertainty of the atmospheric predictors was responsible for the loss of skill with increasing lead time, as shown for CAPE in Sect. 4.3.

## 6 Conclusion

We applied AR-CHaMo for lightning and large hail to the ECMWF reforecasts to develop probabilistic ensemble forecasts for the period 2008–2019 across Europe. In a case study of 15 June 2019, the models provided skilful guidance both for large hail and lightning up to 108 h. The loss of skill with increasing lead time was mostly due to increased spread in CAPE. The predictive skill was quantified in terms of AUC scores as a function of lead time: at short-to-medium ranges hail forecasts are highly skilful ($\geq 0.95$ up to 60 h) and outperform all composite indices. The skill decreases progressively, most rapidly after 108 h in lead time. Some limitations of this study must be considered with the interpretation

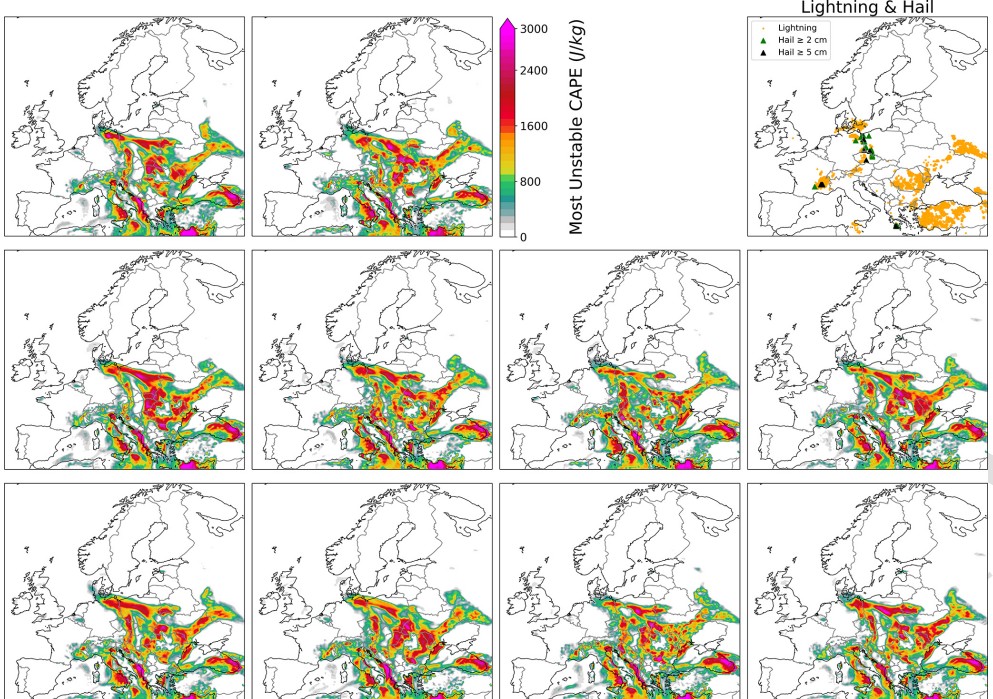

**Figure 9.** Forecast of MUCAPE for 15 June 2019 at 12:00 UTC, initialized at 00:00 UTC on the same day depending on the individual ensemble members. Lightning and hail reports between 12:00 and 15:00 UTC are also shown respectively as yellow dots and triangles (green for hail ≥ 2 cm, black for hail ≥ 5 cm).

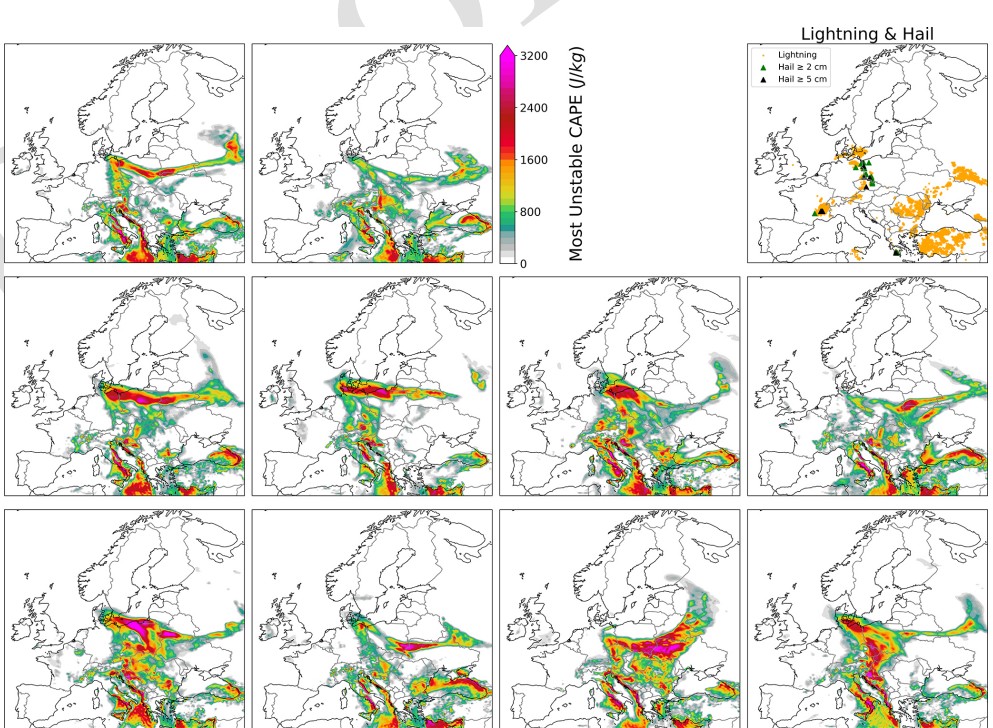

**Figure 10.** As in Fig. 9 for a forecast initialized on 11 June 2019 at 00:00 UTC.

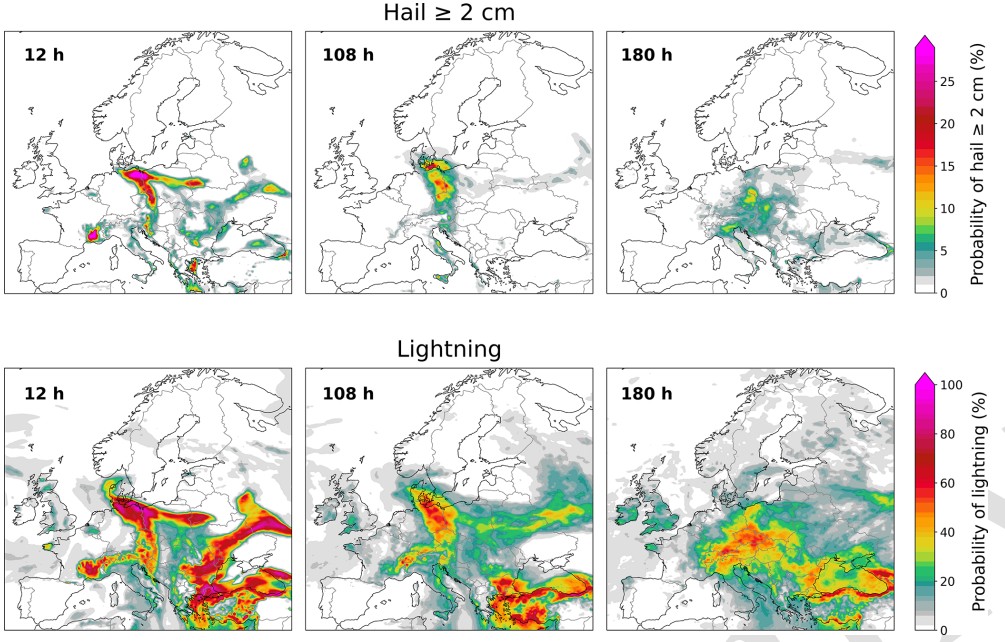

**Figure 11.** Ensemble mean probabilistic forecast of lightning and hail $\geq 2$ cm occurrence for three different lead times ($t - 12$ h, $t - 108$ h and $t - 180$ h).

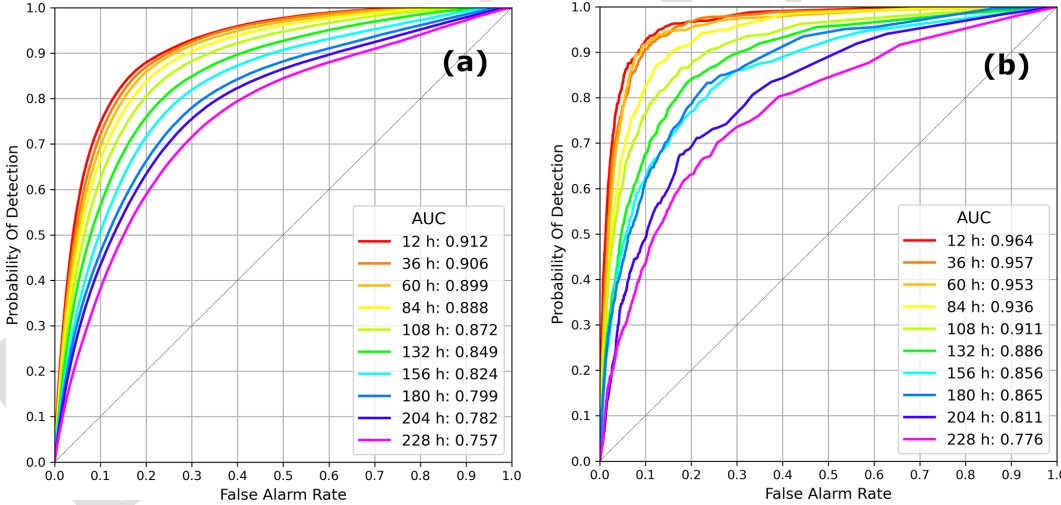

**Figure 12.** Receiver operating characteristic (ROC) curve for **(a)** the hail $\geq 2$ cm and **(b)** the lightning model for lead times from 12 to 228 h. Corresponding values of the area under the ROC curve (AUC) are also shown.

of these results. First, the limited vertical resolution of the ECMWF reforecasts did not allow the exact models developed using ERA5 to be applied, since some of the original predictors could not be calculated due to the limited reforecast data availability. The atmospheric variables selected as predictors in place of the ERA5 variables can only approximate the original predictors and are likely not fully representative of the models developed using ERA5. It is important to note that although the original, more skilful, predictors from ERA5 could not be calculated, the adapted versions still managed to outperform state-of-the-art predictors such as SHP (at short-to-medium lead times) and CAPESHEAR (at all lead times). Another limitation is that forecasts were only available four times a day, at 6 h intervals, due to the limited temporal resolution of ECMWF reforecasts. Given the convective nature of these events it is likely that some hail events for verification were missed in the 6 h intervals between each forecast. To reduce this, forecasts were verified against hail reports in the 3 h following the forecast time.

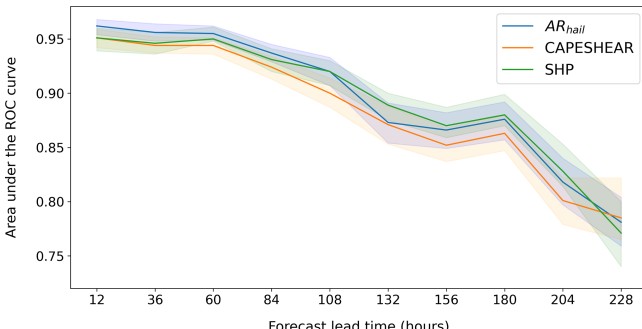

**Figure 13.** AUC depending on forecast lead time for $AR_{hail}$, CAPESHEAR and the significant hail parameter; 1000-member block bootstrap was performed to estimate 95 % AUC confidence intervals for each model (shaded).

Finally, although AR-CHaMo produces probabilities for large hail and lightning, these probabilities are not calibrated. This implies that the computed probabilities do not necessarily coincide with the observed frequency of the phenomena.

Apart from these limitations, the models give valuable guidance on the occurrence of these hazards and represent an improvement compared to state-of-the-art composite parameters in Europe. Future work will involve the application of the full ERA5 models to numerical weather prediction (NWP) to develop hail forecasts operationally and on a pan-European scale. A significant improvement in predictive skill is expected with the use of the most skilful predictors calculated form the ERA5 reanalysis. An extension of this approach to different convective hazards such as severe convective wind gusts and tornadoes can also be foreseen.

## Appendix A: ECMWF forecast sounding for 15 June 2019 at 12:00 UTC (initialized at 00:00 UTC) across central Türkiye

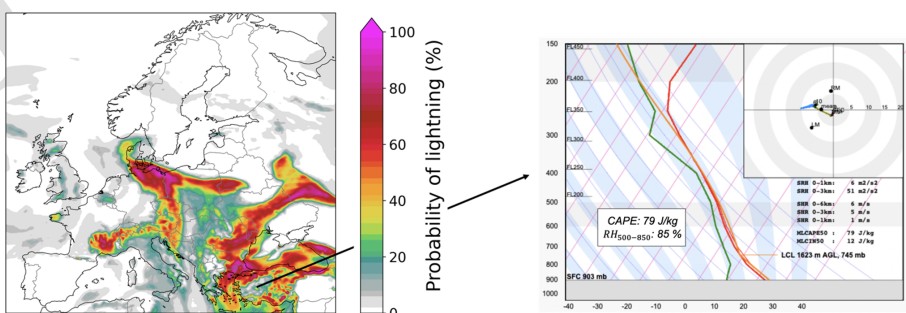

**Figure A1.** Ensemble mean probabilistic forecast of lightning occurrence on 15 June 2019 at 12:00 UTC (initialized at 00:00 UTC). The forecast sounding from the ECMWF deterministic run is shown for central Türkiye. The corresponding CAPE and relative humidity between 500 and 850 hPa ($RH_{500-850}$) are also shown.

**Appendix B:  Lightning and hail forecasts based on SHP and CAPESHEAR**

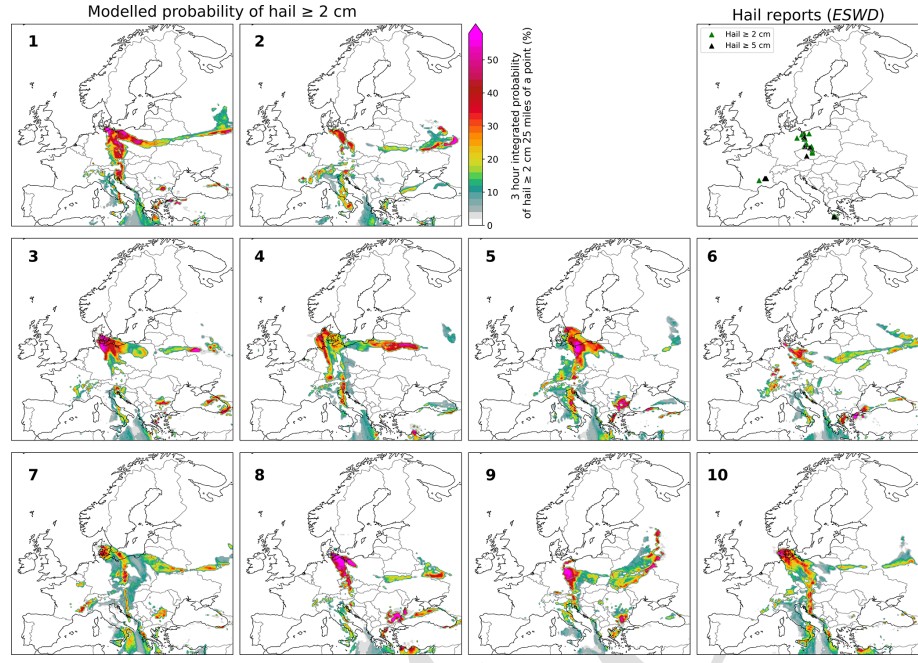

**Figure B1.** Probabilistic ensemble forecast of hail occurrence based on a 1-dimensional logistic SHP model for 15 June 2019 at 12:00 UTC (initialized on 11 June 2019 at 00:00 UTC). Hail reports between 12:00 and 15:00 UTC are shown as triangles (green for hail $\geq 2$ cm but $\leq 5$ cm, black for hail $\geq 5$ cm) in the top-right panel.

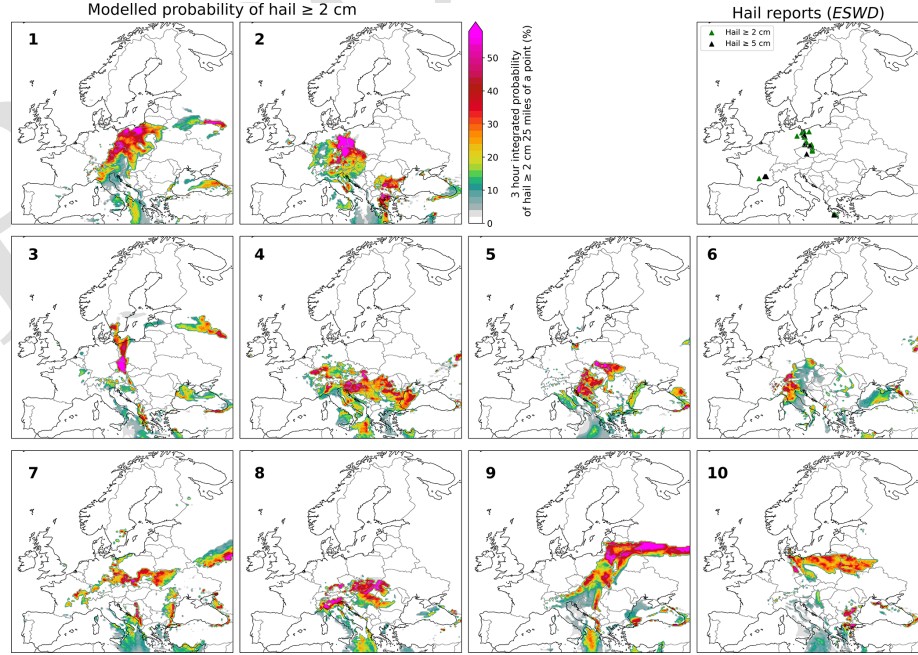

**Figure B2.** As in B1 but for initialization on the 8 June 2019 at 00:00 UTC.

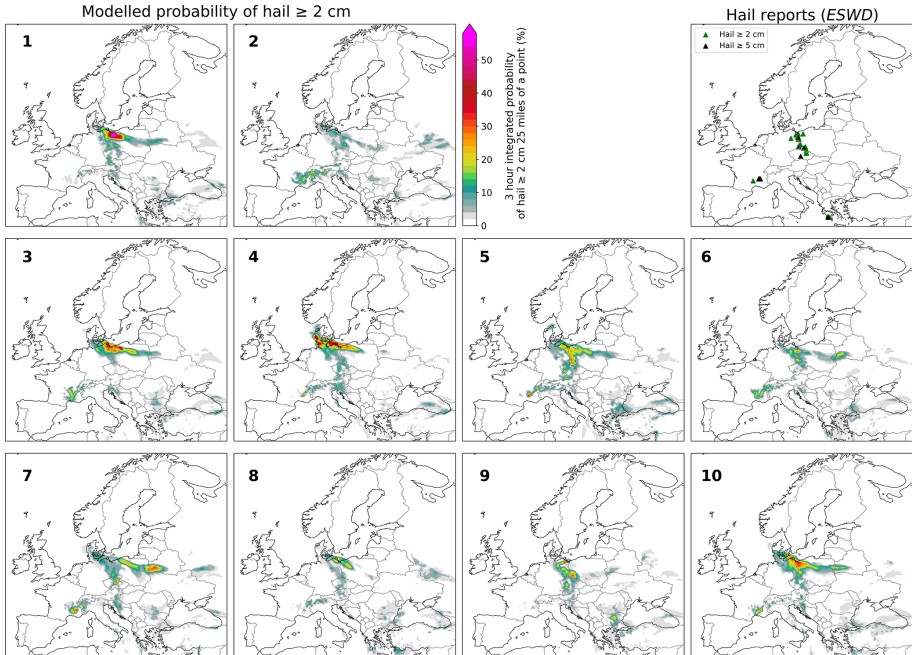

**Figure B3.** Probabilistic ensemble forecast of hail occurrence based on a 1-dimensional logistic CAPESHEAR model for 15 June 2019 at 12:00 UTC (initialized on 11 June 2019 at 00:00 UTC). Hail reports between 12:00 and 15:00 UTC are shown respectively as triangles (green for hail ≥ 2 cm but ≤ 5 cm, black for hail ≥ 5 cm) in the top-right panel.

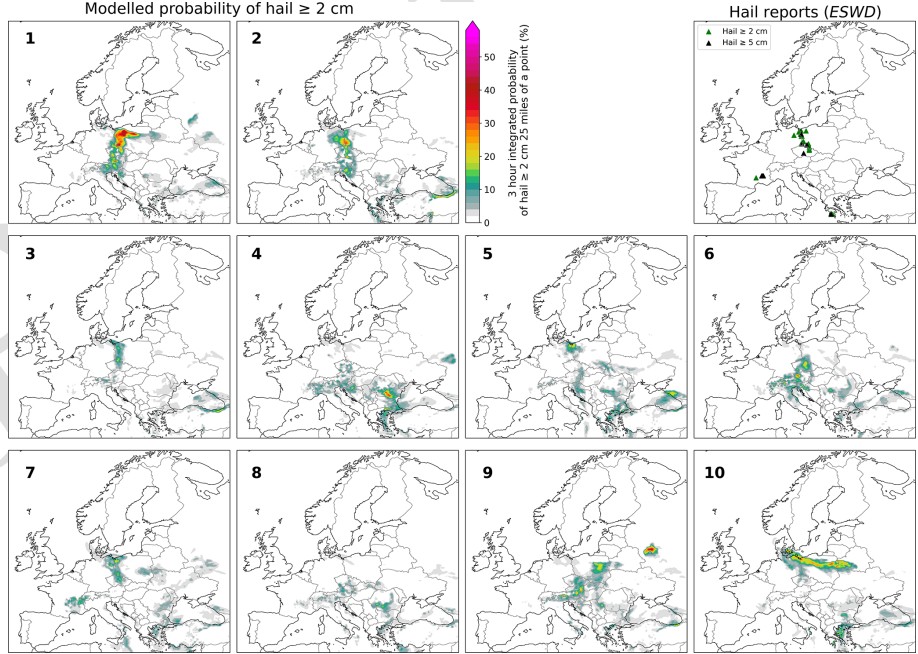

**Figure B4.** As in B3 but for initialization on 8 June 2019 at 00:00 UTC.

**Appendix C: Additional lightning and hail forecasts for 15 June 2019 at 12:00 UTC**

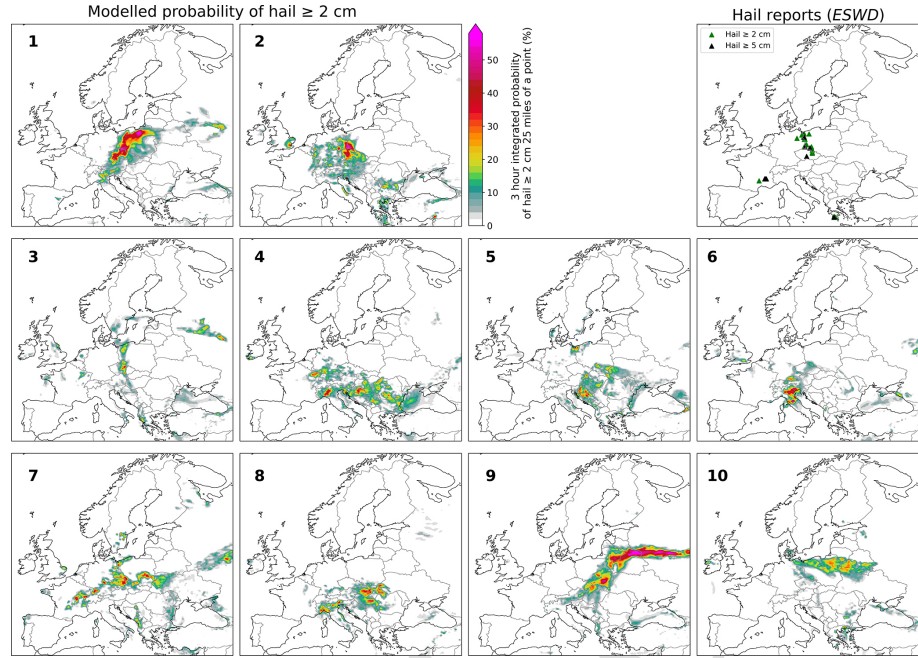

**Figure C1.** Probabilistic forecast of hail occurrence on 15 June 2019 at 12:00 UTC (initialized on 8 June 2019 at 00:00 UTC) for the different ensemble members. Hail reports between 12:00 and 15:00 UTC are shown as triangles (green for hail $\geq$ 2 cm but $\leq$ 5 cm, black for hail $\geq$ 5 cm) in the top-right panel.

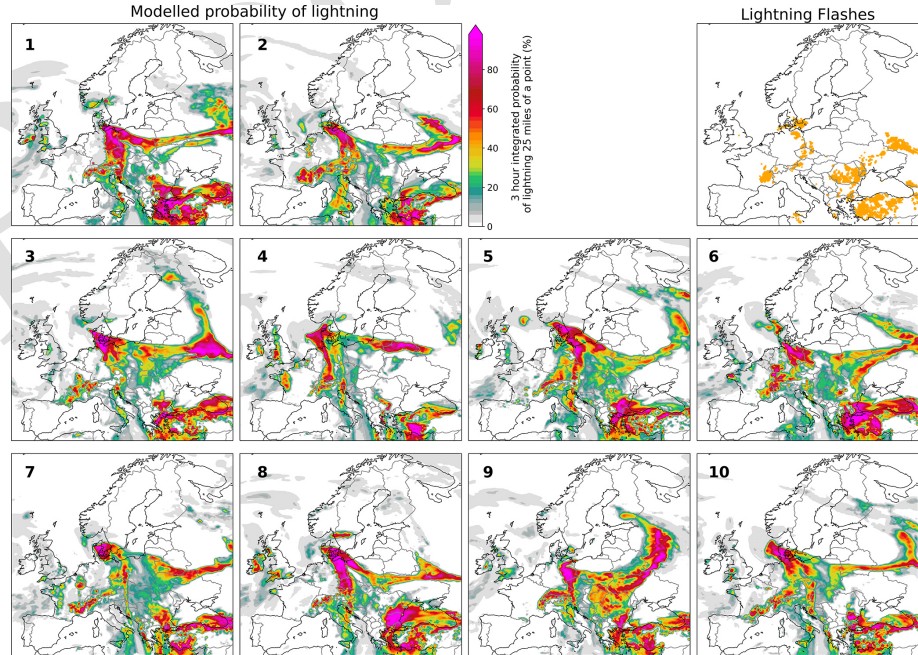

**Figure C2.** Probabilistic forecast of lightning occurrence on 15 June 2019 at 12:00 UTC (initialized on 11 June 2019 at 00:00 UTC) for the different ensemble members. Lightning observations between 12:00 and 15:00 UTC are shown as yellow dots in the top-right panel.

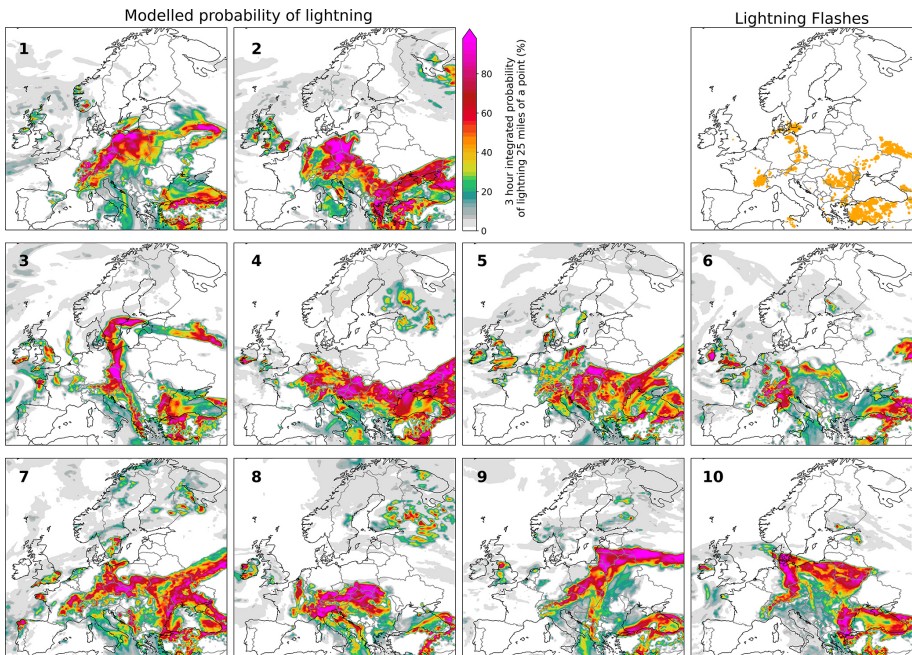

**Figure C3.** As in C2 but for initialization on 8 June 2019 at 00:00 UTC.

## Appendix D: Acronyms of model predictors

**Table D1.** Acronyms for parameters from ERA5 reanalysis used in AR-CHaMo.

| Acronym | Definition |
| --- | --- |
| MU_LI | Most unstable lifted index |
| RH_500–850 hPa | Mean relative humidity between 500 and 850 hPa |
| 1 h Acc. conv. precip. | 1 h Accumulated convective precipitation |
| MU_MIXR | Most unstable mixing ratio |
| MU500_CAPE-10° | Most unstable CAPE (for a parcel originating above 500 m above ground level) released above the $-10\,°C$ isotherm |
| EFF_MU_BS | Effective most unstable bulk shear |
| MU_CAPE | Most unstable CAPE |
| Deep-layer shear | Bulk wind difference between 925 and 500 hPa |

*Code availability.* Code for computing the probabilistic forecasts is available from the corresponding author upon request (francesco.battaglioli@essl.org).

*Data availability.* Probabilistic forecasts are available from the corresponding author upon request (francesco.battaglioli@essl.org). Reforecast data are available through the ECMWF archive (https://www.ecmwf.int/en/forecasts/dataset/operational-archive, ECMWF, 2023).

*Author contributions.* FB developed the code to compute probabilistic lightning and hail forecasts with support from PG. FB analysed the data and produced all the figures. IT extracted the ECMWF reforecast data. FB wrote the manuscript with revising contributions from IT, PG and TP.

*Competing interests.* The contact author has declared that none of the authors has any competing interests.

*Acknowledgements.* We would like to thank and acknowledge ECMWF for providing reforecast data, the Met Office for ATDnet lightning detection data and the many volunteers and ESSL staff for their reports to the ESWD. Mateusz Taszarek is acknowledged to have calculated and provided ERA5 predictors for the original AR-CHaMo models. Matthieu Chevallier is also acknowledged to have provided comments on the manuscript and have contributed to the review process.

*Financial support.* Francesco Battaglioli's contribution to the study was funded by the German Ministry of Education and Research for project 01LP1902G "CHECC", part of the Research Programme "ClimXtreme". Groenemeijer and Púčik's contributions were funded by the Austrian Science Fund (FWF) project P33113-N "PreCAST". Tsonevsky's contribution was funded by ECMWF.

*Review statement.* This paper was edited by Ricardo Trigo and reviewed by two anonymous referees.

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
