# Peer review of "Forecasting Large Hail and Lightning using Additive Logistic Regression Models and the ECMWF Reforecasts"

_Natural Hazards and Earth System Sciences, 2023_

## Author Comment (AC1)

Answer to **RC1**: 'Comment on nhess-2023-40', Anonymous Referee #1, 09 Jun 2023

The manuscript describes the application of AR-ChaMo, being an additive logistic regression model, to forecast large hail and lightning over Europe. The additive logistic regression model predicts the probability of lightning (hence also of a thunderstorm) and the probability of the occurrence of large hail given a thunderstorm forms. The observational data stem from lightning observations and hail reports from the ESWD. The authors show the performance of their model using a case study and also consider the AUROC-score as a verification metric. They also compare their model to two other classical indices and find their model to give equal or better guidance.

The manuscript touches on a very important topic and I find the use of the additive logistic models to predict lightning and hail very interesting. However, I think that the presentation should be improved and some major aspects should be clarified before the manuscript can be accepted.

General comments:
* * *
- Lines 14-15: Where do you show the model's ability to reproduce the climatological distribution and the seasonal cycle?

*Authors: Thank you for pointing this out. The mention of the model's ability to reproduce the climatological distribution and the seasonal cycle refers to results from Battaglioli et al. 2023. For this reason, it was decided to remove these statements from the abstract since they don't belong to the main results of this paper.*

- Section 2.1: I think the model should be described in more detail in this section. Although you refer to another article that should contain all the details, I advocate to repeat the main aspects here to provide the reader a self-contained article. In addition, I have some further remarks/questions

*Authors: The section has been significantly extended to account for the reviewer comment. We expanded on the different training regions for the lightning and hail model, on the rationale behind these areas and on how lightning and hail reports are connected to the corresponding grid points. See next points for a more detailed description.*

(i) Training of the hail-model is based on the hail reports from the ESWD and obviously not all hail occurrences are reported there hence your training data contains many cases where hail occurred but no report was issued. Does this pose a severe problem for the training? How do you mitigate the effect of these non-reported but true hail events?

*Authors: Good point. Indeed not all reports are reported within the ESWD and the underreporting of hail in the dataset is a known limitation (Groenemeijer and Kühne 2014; Taszarek et al. 2020a) especially across certain regions. To minimize the impact of underreporting on the model training, we selected a specific region (Central Europe) and period (2008-2019) as done by Rädler et al. (2018) to train our hail models. For this region/period, the underreporting is the lowest. To best display the different training regions for the lightning and the hail model, we added a figure (Figure 1), adapted from Battaglioli et al. 2023, to the manuscript.*

[Figure]

**Figure 1:** Annual mean distribution of lightning (a) and hail ≥ 2 cm (b) for the period 2008-2019. The black squared in (a) and (b) indicate the training regions for lightning (34.5° – 63.5°N, -9.0° – 46.0°W) and hail ≥ 2 cm (45.0° – 54.0°N, 5.0° – 22.0°W). Adapted from Battaglioli et al. (2023).

*The rationale behind the different training regions for the lightning and the hail models has been added to the text as well: "The model training area for large hail was limited to Central Europe (Fig. 1) where reporting of severe weather in the ESWD is the highest (Groenemeijer and Kühne 2014; Taszarek et al. 2020a). This was done to limit sampling of situations in our training dataset that could have affected the model development, namely when hail occurred but was not reported to the ESWD. The lightning model training region covered a much larger area since the detection efficiency of ATDnet is comparatively stable throughout the domain".*

(ii) If a hail report is within the ESWD, how do you connect the report to the corresponding gridpoints (in space and time)?

*Authors: Thank you for raising this point. We now clarified how reports are connected to the surrounding grid box and associated temporarily with the ERA5 reanalysis. A new paragraph has been added to the manuscript:" Lightning and hail reports were gridded on a 0.25° x 0.25° grid box, the same horizontal resolution of the ERA5 reanalysis. In addition, the time of reports was rounded down to the nearest full hour. This allowed to best sample the pre-convective environment associated with each lightning observation or hail report as done by Rädler et al. (2018) and Taszarek et al. (2021b).".*

(iii) Judging from the figures in other sections, you apply AR-ChaMo per ensemble member. This should be stated explicitly.

*Authors: Yes we do so. See section (v) of the review document.*

(iv) Given that you also refer to a "Ensemble mean probabilistic forecast" (e.g. captions of figures 3 and 4 and also line 51 or line 170), how do you compute this mean forecast? Is it the mean of the forecasted probabilities of AR-ChaMo on the individual ensemble members?

*Authors: Yes it is. This has now been clarified at the beginning of section 5 in the manuscript.*

(v) What exactly would your forecast-product look like, i.e. what should a forecaster see? Do you want to forecast a probability per member or a single probability that is a combined/mean probability derived from all ensemble members? If it is the latter, I suggest to add a figure of that combined probability.

*Authors: The purpose of this section is exclusively to present the AR-CHaMo as the conceptual base for our probabilistic forecasts, its framework and the way it was developed. The application of the model (to different ensemble members or to the ensemble mean) is presented in Section 3 and 4 where we add more explanation regarding the points being raised here e.g., applying it to each ensemble member. In section 3 and 4 we display both the forecast per ensemble member and the ensemble mean.*

- Section 2.2: Does the ECMWF model change from 46r1 to 47r1 (as mentioned in line 71) have an impact on the training of AR-ChaMo?

*Authors: No we don't expect this to have had an impact on training or application of AR-CHaMo. To clarify in the text we added the following: "Such change in IFS cycle is not expected to have caused discontinuities in convective parameters calculation (e.g., in CAPE) since the convection scheme was not modified from 46r1 to 47r1."*

- Section 2.3: How many hail reports are within your trainingset?

*Authors: Good point! We added a sentence mentioning the amount of lightning and hail reports: "In total, 10872890 lightning observations and 5493 reports of large hail were considered within the verification dataset."*

- Section 3:

(i) Why do you adapt the model based on ERA5 with this complicated process and not retrain it on the reforecast-data?

*Authors: That is not possible, since reforecasts are run only twice a week. To sample the conditions associated with large hail well, we would need a forecast time close to the*

*analysis time, and the forecasts are available only every 6 hours. This leaves very few times at which we would have both a reasonably accurate and temporally nearby estimate of the state of the atmosphere to any hail report, and definitely too few cases to train the model on.*

*A paragraph detailing these limitations and the reasons for the chosen approach has been added to the Section.*

(ii) Where do all these adaptation-formulas come from, i.e. how are they derived?

*Authors: Thank you for pointing this out. We clarified this in the text: "following Eq. (3), based on the assumption that each hour probability is independent of that in another hour". Eq. 3 simply regards the probability of each hour being independent of that in another hour. We know that this is not necessarily true, but we wanted to keep the approach simple rather than additionally considering the correlations between probabilities between adjacent hours.*

*With Eq. 3 we want to transform the 1 hour probability output from the model to a 3 hour equivalent. To clarify this, let's take an example: If we have 10% chance in the first hour and 10% in a second hour, we can have these possibilities:*

*event in first hour and event in second hour: P = 0.1 * 0.1 = 0.01. Event in first hour but not in second hour P = 0.1 x 0.9 = 0.09. Event in second hour but not in first hour P = 0.9 x 0.1 = 0.09. Event not in first and not in second hour: P = 0.9 x 0.9 = 0.81. So, chance of one or more events = 0.01 + 0.09 + 0.09 = 0.19. The chance of zero events = 0.81. The formula generalized this to three probabilities.*

*The same concept applies for Eq. 2 with the difference that instead of the hourly periods it uses the ratio of the surface areas.*

(iii) Please clearly state the quantities that your lightning model and your hail model try to forecast, e.g.: The probability of occurrence of at least two lightnings within 3 hours and an area of 0.2°x0.2°.

*Authors: We modified the last sentence of Section 3 to the following: "With these adaptations, we applied the AR-CHaMo based on ERA5 to ECMWF reforecasts yielding 3 hourly ensemble lightning and hail probabilistic forecasts at 0.2° x 0.2° spatial resolution for the period 2008-2019 and the whole of Europe" adding mentions about the probabilistic nature of the forecast, the temporal and spatial resolution.*

- Line 96: You state that the probabilities are calibrated. Please add some reliability plots.

*Authors: Thank you for pointing this out. "Calibrated" is indeed the wrong choice of word here. No calibration of modeled output to the observations was performed here. For "calibrated" here we mean that the probabilities had to be adapted from an ERA5 grid*

*(0.25° x 0.25°) to an ECMWF reforecast one (0.2° x 0.2°). In this regard, please refer to equation 2. To clarify this, we modified "calibrated" to "adapted".*

- Line 108: Did you also upscale the observations?

*Authors: The observations were also accumulated at 3 hourly intervals for verification purposes. The procedure is explained at lines 172-175: "Ensemble mean probabilistic forecasts were systematically verified against … hail reports from the ESWD in the three hours following the forecast time (00– 03UTC, 06–09UTC, 12–15UTC, 18–21UTC).*

- Line 123: How did you upscale the probabilities?

*Authors: The probabilities were upscaled using equation 2 but instead of an ECMWF and ERA5 grid size we converted the ECMWF grid size to the equivalent of a 25 miles grid box. Effectively the forecasts were adapted to an ECMWF grid (from an ERA5 one) and then upscaled to yield the equivalent probability over a 25 miles grid. To clarify this in the text we added a reference to Equation 2.*

- Lines 166-167: To be more precise, you showed that for this case the forecast was skilful up to 108h in advance but a single case study cannot support such a conclusion in general.

*Authors: Thank you for the comment. Indeed we agree one case study is not enough to say that our forecasts are skillful at a certain lead time. With lines 166-167 we want to point out that "**For** this case study … the logistic model provided a skillful forecast up to 108 hours", not that the model is always skillful up to 108 hours. To provide a more objective and general validation of the AR-CHaMo models we systematically evaluated their performance throughout the whole period 2008-2019 using AUC scores in Section 5.*

*To highlight the fact that we are not drawing conclusions on the performance of the model only on the base of a case study, a sentence at the beginning of Section 5 has been added to the manuscript: "In order to provide an evaluation of the model's skill not tied to a single case study, the ensemble mean probabilistic forecasts were systematically verified …"*

- Section 5: Please add a few sentences that clarify the quality aspect that is covered by the AUROC-score.

*Authors: Yes, good point. We added a sentence describing the role of the AUC scores in our study, namely measuring: "the ability of the hail (lightning) model to discriminate between hail (lightning) and non-hail (non-lightning) situations. AUC values can range from 0 to 1: a model that has an AUC of 0.5 has the same skill of random guessing, the closer the AUC is to 1 the better the model can classify observations into classes. More specifically, an AUC of 0.8 to 0.9 is considered excellent, and more than 0.9 is considered outstanding (Hosmer et al. 2013). "*

- Line 172: In section 2.3 you define a lightning case as a one-hour period. Now you consider a three-hour period. What is the final product? Mixing of both products is misleading.

*Authors: Thank you for the comment. Indeed there is a lack of consistency between section 2.3 and line 172. A lightning case is defined as a three-hour period with at least two lightning strikes per grid box. This has now been modified in the text in section 2.3.*

- Line 183: Are the 1-dimensional logistic regression models trained on the same trainingset (where the appropriate quantities are computed) as AR-ChaMo?

*Authors: Yes, they were trained on the same dataset. To clarify this in the text we added a sentence to this section: "The performance of ARhail was compared with that of 1-dimensional logistic models trained using CAPE-shear and SHP as predictors and the same training dataset as for ARhail".*

- Appendix B: Maybe I missed it but I think there is no reference to appendix B suggesting that this appendix is not needed.

*Authors: There is a reference to Appendix B in lines 164-165: "The T-108-hour lightning forecast (Appendix B) was also in good agreement with the T-12-hour forecast"*

- Appendic C: The caption of the table says that this is the list of parameters that enter AR-ChaMo based on ERA5. I suggest to extend that table to also give a list of parameters that enter the AR-ChaMo model used in present study.

*Authors: Good point. The new predictors that entered AR-CHaMo model used in the present study are: Deep Layer Shear, Specific humidity at 925 and Wet Bulb Zero height. We have added them to the table. Since specific humidity and wet bulb are explicitly mentioned in the text without acronyms they were not included in Appendix C. Deep Layer Shear was included with its definition: "Bulk wind difference between 925 and 500 hPa".*

**Appendix C - Acronyms of model predictors**

| Short name | Long name |
|---|---|
| MU_LI | Most Unstable Lifted Index |
| RH_500–850hPa | Mean Relative Humidity between 500 and 850 hPa |
| 1h Acc. Conv. Precip. | 1 hour Accumulated Convective Precipitation |
| MU_MIXR | Most Unstable Mixing Ratio |
| MU500_CAPE-10° | Most Unstable CAPE (for a parcel originating above 500 m AGL) released above the -10°C isotherm |
| EFF_MU_BS | Effective Most Unstable Bulk Shear |
| MU_CAPE | Most Unstable CAPE |
| Deep Layer Shear | Bulk wind difference between 925 and 500 hPa |

- Figures 1, 4, 5: I suggest to add a panel that contains the guidance from the other composite indices for comparison.

*Authors: Thank you for pointing this out. To take this comment in consideration we produced forecasts for the 15th of June 2019 and three initialisation times (T-12h, T-108h, T-180h) - as for AR-CHaMo - but for the 1-dimensional logistic models based on SHP and CAPESHEAR. The 12 hour forecasts plots (now Figure 3 and Figure 4) have been added to the main text along with a discussion of the main differences with the AR-CHaMo output in Section 4.1: "To compare the AR-CHaMo forecast with that of existing composite parameters, we produced probabilistic hail forecasts for the same timestep and initialization time based on two 1-dimensional logistic models trained using SHP (Figure 3) and CAPESHEAR (Figure 4). The SHP model is in agreement regarding the Germany-Poland-Czechia and south-eastern France regions, but compared to AR-CHaMo, yields high hail probabilities also across regions where no hail was reported e.g., the Balkans and Eastern Europe. The CAPESHEAR model, on the other hand, identifies well the south-eastern France region but places the highest probability of hail across northern Germany and northern Poland away from the highest density of hail reports to the south." The plots for the longer lead times have been added to the Appendix.*

[Figure]

**Figure 3: Probabilistic ensemble forecast of hail occurrence based on a 1D logistic SHP model for the 15th of June 2019 at 12:00 UTC (initialized on the 15th of June 2019 at 00:00 UTC). Hail reports between 12:00 UTC and 15:00 UTC are shown as triangles (green for hail ≥ 2 cm but ≤ 5 cm, black for hail ≥ 5 cm) in the right-top panel.**

[Figure]

**Figure 4: Probabilistic ensemble forecast of hail occurrence based on a 1D logistic CAPESHEAR model for the 15th of June 2019 at 12:00 UTC (initialized on the 15th of June 2019 at 00:00 UTC). Hail reports between 12:00 UTC and 15:00 UTC are respectively as triangles (green for hail ≥ 2 cm but ≤ 5 cm, black for hail ≥ 5 cm) in the right-top panel.**

Technical comments:
* * *
- Lines 66, 89: I think it should read "Table C1"?

*Authors: Thank you for pointing this out. The reference to Table 1 is correct. We noticed that Table 1 was not correctly copied in the preprint, hence the confusion about the reference. The comment was addressed by adding Table 1 in the text.*

- Line 88: It should read "Battaglioli et al. (2023)".

*Authors: Thank you. I added the parentheses.*

- Lines 92, 94: There is no Table 2.

*Authors: Thank you for pointing this out. For some reason both Table 1 and Table 2 did not make it in the preprint. Both tables have been added at the bottom of the manuscript before Figure 1.*

- Line 129: It should read "...France. This...".

*Authors: Yes, thank you for pointing this out. I added a full stop between "France" and "This".*

- Line 138, 139: It should read: "...probabilities for lightning...".

*Authors: Modified to "probabilities of lightning".*

- Line 191: I think it should read "Section 4.3"?

*Authors: Yes indeed, it should read "Section 4.3". This has been corrected in the text.*

- Lines 194, 197: Prior to this section you used the British version "skilful" instead of "skillful".

*Authors: True, in order to keep consistency, all "skillful" in text have been modified to "skilful".*

- Line 194: I think it should read "based on the period" instead of "for the period".

*Authors: In this case we mean "for the period" meaning that we are applying the models to all available ensemble forecasts between 2008 and 2019. The comment probably refers to the fact that the logistic models were also trained "based on the period" 2008-2019 but in this case that is not what we are referring to. For this reason, we left "for the period" in the text.*

**References:**

1. Groenemeijer, P., and T. Kühne, 2014: A Climatology of Tornadoes in Europe: Results from the European Severe Weather Database.Mon. Wea. Rev., 142, 4775-4790, doi:10.1175/mwr-d-14-00107.1.

2. Rädler, A. T., P. Groenemeijer, E. Faust, and R. Sausen, 2018: Detecting Severe Weather Trends Using an Additive Regressive Convective Hazard Model (AR-CHaMo). J. Appl. Meteor. Climatol., 57, 569–587, https://doi.org/10.1175/JAMC-D-17-0132.1

3. Taszarek, M., J. Allen, P. Groenemeijer, R. Edwards, H. Brooks, V. Chmielewski, and S. Enno, 2020(a): Severe Convective Storms across Europe and the United States. Part I: Climatology of Lightning, Large Hail, Severe Wind, and Tornadoes. J. Climate, 33, 10239- 10261, doi:10.1175/jcli-d-20-0345.1.

4. Taszarek, M., N. Pilguj, J. Allen, V. Gensini, H. Brooks, and P. Szuster, 2021b: Comparison of convective parameters derived from ERA5 and MERRA2 with rawinsonde data over Europe and North America. J. Climate, 1-55, doi:10.1175/jcli-d-20-0484.1.

---

## Author Comment (AC2)

**1 Overview**

In this manuscript, the authors present the performance of an additive logistic regression model called "AR-ChaMo" that is able to predict the probability of lightning and hail occurrence. AR-ChaMO model for hail is compared with two documented convective indices and results show that AR-ChaMo outperforms them.
While the manuscript is well-written and the presented results are very interesting, and of great importance, I would suggest some major revisions to be made before the manuscript is accepted for publication.

**2 Major comments**

· The study is based on the additive logistic models developed by Rädler et al. (2018) but they were based on the much coarser ERA-Interim reanalysis dataset and in the current manuscript, in Section 3, it mentioned that ERA5 was used to train the models with additional parameters. While in Rädler et al. (2018) it reads in their Section 6 "*This model may be improved further by using additional predictor parameters. Such parameters may include low-level moisture, lapse rates, lifted condensation level, or height of the melting level.*", it seems that the authors of the current manuscript have taken into account this suggestion but they do not present their calculations. In addition, Battaglioli et al. (2023) study has not been published yet, and several aspects of the approached used are not currently accessible. The table with some acronyms of the parameters used is not sufficient in order to publish a method and make it reproducible by the scientific community.

*Authors: Thank you for raising this point. We understand that the current paper is heavily based on Battaglioli et al. 2023 (Journal of Applied Meteorology and Climatology). The JAMC paper has been conditionally accepted and is now undergoing the last round of reviews. For reasons we are not aware of, the review process has been extremely slow (6 months between submission and receiving the reviews), this has caused delays in the publication of Battaglioli et al. 2023.*

*We have decided to upload the manuscript as a preprint. It can be accessed at the following link: https://www.preprints.org/manuscript/202308.0314/v1*

*The Battaglioli et al. 2023 paper presents a thorough description of the model selection procedure that yielded the final lightning, hail > 2 cm and hail > 5 cm models out of 172 possible predictors. Nonetheless, to account for the reviewer's comment, an addition to the section 2.1 paragraph has been included citing the different metrics and procedure used for model selection: "model selection procedure based on an ingredients-based approach (Doswell et al. 1996), the Deviance Explained (Wood 2006) and the Bayesian Information Criterion (BIC, Schwarz 1978) scores. Out of 172 available parameters from the ERA5 reanalysis, the model selection procedure with*

*Deviance Explained (the higher, the better) and BIC (the lower, the better) yielded a 5-dimensional lightning model and a 4-dimensional conditional hail model".*

*With this addition we aim to provide the reader with a clearer view of the metrics and the procedure used for model selection. Rather than repeating the whole procedure in detail, the reader is directed to the Battaglioli et al. 2023 manuscript (now available).*

While the study is dealing with both hail and lightning forecasting, the latter is not discussed at all in the Introduction. Please expand this section and use more references for lightning forecasting. Ideally, other approaches for lightning forecasting should be discussed and compared, similar to hail.

*Authors: Thank you for pointing this out. The introduction has now been heavily edited to consider the reviewer's comment. A description of lightning forecasting techniques in the literature has been added and 18 references to lightning-related papers have been added to the section. We now believe the introduction describes both lightning and hail thoroughly. For a more detailed view of the edits, we refer to the updated manuscript.*

· The Significant Hail Parameter (SHP) is able to distinguish environments with very large hail (>=5 cm) and with small hail. How did you compare your model's result with SHP given that you used ESWD data with hail >=2 cm? Please consider adding a few more graphs either in main text or as an appendix to visually compare examples for large hail forecasts with SHP, CAPESHEAR, and AR-ChaMo.

*Authors: Thank you for raising this point. Indeed SHP is meant to delineate very large hail to large hail. Nonetheless SHP represents a skillful predictor for large hail in Europe (Czernecki et al. 2019). For this reason, it was selected as a possible composite parameter to compare with ARhail. Other parameters e.g., the Large Hail Parameter (LHP) were considered for testing, but the limited vertical resolution of the ECMWF reforecast data made it impossible to calculate.*

*To address the second part of the comment we produced forecasts for the 15th of June 2019 and three initialization times (T-12h, T-108h, T-180h) - as for AR-CHaMo - but for the 1-dimensional logistic models based on SHP and CAPESHEAR. The 12 hour forecasts plots (now Figure 3 and Figure 4) have been added to the main text along with a discussion of the main differences with the AR-CHaMo output in Section 4.1: "To compare the AR-CHaMo forecast with that of existing composite parameters, we produced probabilistic hail forecasts for the same timestep and initialization time based on two 1-dimensional logistic models trained using SHP (Figure 3) and CAPESHEAR (Figure 4). The SHP model is in agreement regarding the Germany-Poland-Czechia and south-eastern France regions, but compared to AR-CHaMo, yields high hail probabilities also across regions where no hail was reported e.g., the Balkans*

*and Eastern Europe. The CAPESHEAR model, on the other hand, identifies well the south-eastern France region but places the highest probability of hail across northern Germany and northern Poland away from the highest density of hail reports to the south. "*

*The plots for the longer lead times have been included in the manuscript, in the Appendix.*

[Figure]

*Figure 3: Probabilistic forecast of hail 2 cm occurrence on the 15th of June 2019 at 12:00 UTC (initialized on the 15th of June 2019 at 00:00 UTC) for the individual ensemble members. Hail reports between 12:00 UTC and 15:00 UTC are shown as triangles (green for hail 2 cm but 5 cm, black for hail 5 cm) in the right-top panel.*

[Figure]

*Figure 4: Probabilistic ensemble forecast of hail occurrence based on a 1D logistic SHP model for the 15th of June 2019 at 12:00 UTC (initialized on the 15th of June 2019 at 00:00 UTC). Hail reports between 12:00 UTC and 15:00 UTC are shown as triangles (green for hail 2 cm but 5 cm, black for hail 5 cm) in the right-top panel.*

**3 Minor comments**

1. Acronyms are used without explicitly defining them, even in the Abstract (i.e., ECMWF, CAPE, etc).

   *Authors: We have added the definition before the acronyms in the Abstract and throughout the text.*

2. Wet Bulb Zero Height is mentioned in the Abstract but it is not mentioned anywhere else. Was it used?

   *Authors: In Table 2 we changed 0° height with Wet Bulb Zero Height since the latter was the one used in the final model.*

3. In Line 66 Table 1 should be Table 1c.

   *Authors: Thank you for the comment. As mentioned already for reviewer #1, Table 1 is correctly mentioned here. The problem is that Table 1 was not correctly copied in the Preprint. Now this has been corrected.*

4. Rädler et al. (2019) should be Rädler et al. (2018).

*Authors: Yes, good point. This has been changed accordingly.*

5. Lighting should be lightning in Lines 80 and 171.

   *Authors: Thank you, we corrected "lighting" to "lightning" throughout the text.*

6. I am not totally sure, but the time convention is 00z or 0000 UTC or 00:00 UTC, and not 00 UTC as it is used now. Please double check and modify accordingly.

   *Authors: Thank you for pointing this out. We changed all time instances to the format "00:00 UTC".*

7. In Line 135: J/kg – J kg$^{-1}$

   *Authors: Changed accordingly.*

**References:**

1. Rädler, A. T., P. Groenemeijer, E. Faust, and R. Sausen, 2018: Detecting Severe Weather Trends Using an Additive Regressive Convective Hazard Model (AR-CHaMo). J. Appl. Meteor. Climatol., 57, 569–587, https://doi.org/10.1175/JAMC-D-17-0132.1
2. Battaglioli, F., Groenemeijer, P., Púčik, T., Taszarek, M., Ulbrich, U., and H. Rust, 2023: Logistic modelling of (very) large 315 hail occurrence in Europe and the United States (1950-2021)

3. Czernecki, B., M. Taszarek, M. Marosz, M.Półrolniczak, L. Kolendowicz, A. Wyszogrod zki, and J. Szturc, 2019: Application of machine learning to large hail prediction—The importance of radar reflectivity, lightning occurrence and convective parameters derived from ERA5. Atmos. Res., 227, 249–262, https://doi.org/10.1016/j.atmosres.2019.05.

---

## Author Response (AR2)

Answer to **RC1**: 'Comment on nhess-2023-40', Anonymous Referee #2, 9 Sep 2023

\*\*\* General comments \*\*\*

(1) Although the authors clarified that they do not claim to have developed a calibrated model, I think that this fact should be clearly stated in the text. Hence I suggest to add a sentence in Section 5 and/or Section 6 like "Although AR-ChaMo produces probabilities for large hail and lightning, these probabilities are not calibrated. This implies that the computed probabilities do not necessarily coincide with the observed frequency of the phenomena but the model might nevertheless give a valuable guidance on occurrence of these hazards."

*Authors: Thank you for pointing this out. We added this point to Section 6: "Finally, although AR-CHaMo produces probabilities for large hail and lightning, these probabilities are not calibrated. This implies that the computed probabilities do not necessarily coincide with the observed frequency of the phenomena.*

*Apart from these limitations, the models give a valuable guidance on occurrence of these hazards and represent an improvement compared to state-of-the-art composite parameters in Europe."*

(2) I appreciate the interpretation of the meteorological situation at the end of Section 4.1 and in Section 4.2, however one should be careful to explain the behaviour of an ensemble by analyzing the deterministic run (or a singe ensemble member). The issue here is that each ensemble members has by definition its own realization and a vertical profile might look quite different between the members. In my opinion this should be communicated to the reader to provide a word of caution with this sort of analysis. A more probabilistic way of doing the same analysis would be to analyse the mean or median profile together with an indication of the ensemble spread of the profiles (e.g. by plotting the standard deviation or a difference of quantiles).

*Authors: We clarified this in the text in Section 4.2: "Although this analysis provides a physical explanation of the different lightning probabilities, it is important to note that here we considered vertical profiles from the two deterministic runs only while the profiles of the single ensemble members could differ significantly, for instance in the realization of mid-level moisture, to the deterministic ones"*

\*\*\* Specific comments \*\*\*

Line 83:
I think it should read "grid" instead of "grid box".

*Authors: Adapted accordingly.*

Line 115:
Maybe it should read "grid" instead of "box"?

*Authors: Adapted accordingly.*

Lines 131 to 134:
Indeed, but a common strategy is to train separate models per leadtime. From this sentence I think that you use a single model and apply it for all leadtimes. Please add a sentence that clarifies that to the reader.

*Authors: The sentence has been rephrased to the following: "By training a single model for all lead times, if for instance two hail reports occurred on Tuesday at 12:00 UTC and on Wednesday at 18:00 UTC, it would not be possible to qualitatively compare the environmental conditions at these two time steps. This is because for a forecast initialized on Tuesday at 00 UTC, the environmental conditions associated with the Wednesday at 18:00 UTC report would be subject to larger uncertainty due to the larger lead time.*